# DSP-KD: Dual-Stage Progressive Knowledge Distillation for Skin Disease Classification

**DOI:** 10.3390/bioengineering11010070

**Published:** 2024-01-10

**Authors:** Xinyi Zeng, Zhanlin Ji, Haiyang Zhang, Rui Chen, Qinping Liao, Jingkun Wang, Tao Lyu, Li Zhao

**Affiliations:** 1Hebei Key Laboratory of Industrial Intelligent Perception, College of Artificial Intelligence, North China University of Science and Technology, Tangshan 063210, China; zengxinyi@stu.ncst.edu.cn (X.Z.); zhanlin.ji@ncst.edu.cn (Z.J.); 2Telecommunications Research Centre (TRC), University of Limerick, V94 T9PX Limerick, Ireland; 3Department of Computing, Xi’an Jiaotong-Liverpool University, Suzhou 215000, China; haiyang.zhang@xjtlu.edu.cn; 4Changgeng Hospital, Institute for Precision Medicine, Tsinghua University, Beijing 100084, China; cra01052@btch.edu.cn (R.C.); lqpa00594@btch.edu.cn (Q.L.); 5Beijing National Research Center for Information Science and Technology, Institute for Precision Medicine, Tsinghua University, Beijing 100084, China; wangjk0123@163.com

**Keywords:** skin disease classification, knowledge distillation, diverse knowledge, ISIC2019

## Abstract

The increasing global demand for skin disease diagnostics emphasizes the urgent need for advancements in AI-assisted diagnostic technologies for dermatoscopic images. In current practical medical systems, the primary challenge is balancing lightweight models with accurate image analysis to address constraints like limited storage and computational costs. While knowledge distillation methods hold immense potential in healthcare applications, related research on multi-class skin disease tasks is scarce. To bridge this gap, our study introduces an enhanced multi-source knowledge fusion distillation framework, termed DSP-KD, which improves knowledge transfer in a dual-stage progressive distillation approach to maximize mutual information between teacher and student representations. The experimental results highlight the superior performance of our distilled ShuffleNetV2 on both the ISIC2019 dataset and our private skin disorders dataset. Compared to other state-of-the-art distillation methods using diverse knowledge sources, the DSP-KD demonstrates remarkable effectiveness with a smaller computational burden.

## 1. Introduction

Skin diseases rank as the fourth leading non-fatal burden cause worldwide, affecting around 1.9 billion people and contributing to a global total of 5.5 million disability-adjusted life years (DALY) [1]. Despite their low fatality rate and high curability, if not diagnosed and treated early, skin diseases can negatively impact individual lives and family harmony by causing physical disability, psychosocial problems, and long-term distress. Moreover, they may progress to the most threatening and least predictable forms of skin cancer at their advanced stages [2]. Due to the variety of skin diseases, many presenting with a similar symptomatology, which makes the task of characterizing them complex and time-consuming, even experienced dermatologists are only able to achieve about 75% accuracy when using dermoscopy to diagnose melanoma [3].

In the last few years, researchers have conducted numerous studies to alleviate the escalating demand for dermatosis clinical diagnostics [2], aiming to achieve the accurate and impartial diagnosis of skin conditions. To this end, they have applied deep learning techniques to the analysis of dermatoscopic images. This integration has resulted in a significantly higher disease detection rate compared to traditional unaided clinical diagnosis methods such as the seven-point checklist [4] and the ABCDE Rule (asymmetry, irregular border, multiple colors, diameter, enlarging lesion) [5]. Convolutional neural networks (CNNs) stand out as one of the most prevailing deep learning approaches. Inspired by how our eyes work, they extract advanced features from images by transitioning from raw images to an N-dimensional vector through a series of convolutional calculations, with N serving as a hyperparameter. These acquired abstract features enable CNNs to capture vital information more efficiently, boosting accuracy in less time [6,7]. Additionally, they adeptly address various forms of skin disease characteristics, effectively mitigating the limitations in conventional manual feature extraction methods that fall short in dealing with complex skin conditions. However, the model parameters also become increasingly extensive with the ongoing advancement and heightened performance of CNNs. Deploying these heavy-duty deep learning models to clinical medical devices with limited resources poses a growing challenge [8].

In response to this scenario, researchers are shifting their focus toward lightweight algorithms aimed at building effective healthcare systems. Hinton et al. introduced a model compression and acceleration strategy known as Knowledge Distillation (KD). This approach involves transferring optimal parameter values tailored for a specific task from a complex large model (teacher network) to a deployable small model (student network) through training, a process referred to as “distillation” [9]. The student model attains performance comparable to that of the large model based on the knowledge acquired from the teacher, yet without a significant increase in its own parameter count. KD has found widespread application and diverse branches in the optimization of deep learning models. In the context of image classification tasks, the transfer of knowledge is typically achieved through either the logits or features information obtained from the teacher [10]. As illustrated in Figure 1, in Logit-Based Knowledge Distillation, the knowledge acquired by the student model is primarily manifested in the probability distribution at the output layer of the model. On the other hand, Feature-Based Knowledge Distillation focuses on transmitting intermediate-layer features from the teacher model, prompting the student model to learn how to generate similar feature representations when processing input data.

In recent years, feature-based KD methods have dominated performance [11]. Chen et al. posited that feature maps offer richer information as opposed to logits. Specifically, they identify advanced semantic features in input images as the key to overcoming performance bottlenecks in classification tasks [12]. Conversely, Hsu et al. argued that the developmental potential of traditional KD methods has been overlooked. Given the definition of the same optimization standards, logits are typically a more effective source of knowledge [13]. Since pure logit distillation is challenging to learn and sufficient feature dimensionality is crucial for model design, what about combining them within an optimal framework to maximize the mutual information between the teacher and student representations? With this hypothesis in mind, we conducted an extensive review and experimentation within the knowledge distillation domain. Ultimately, we devised a dual-stage progressive knowledge distillation framework for the skin disease classification task, incorporating distinct knowledge sources. Our method integrates an improved feature-based mask generation distillation strategy and normalized logit knowledge distillation, aiming to enhance the use of lightweight deep learning models in skin disease classification.

KD methods in medical embedded systems hold immense potential. However, to the best of our knowledge, there is currently limited in-depth and widespread exploration of improved KD algorithms for multi-class skin disease diagnosis. This study aims to address the challenges of applying KD methods in the field of skin disease classification. The main contribution of this paper can be summarized as follows:We propose a novel progressive fusion distillation framework with enhanced diverse knowledge, based on dermatoscopic images. Unlike other approaches employing simultaneous multi-teacher distillation, our framework adopts a dual-stage distillation paradigm with the same teacher. This strategy gradually minimizes teacher–student differences’ adverse impact on performance and maximizes learning “from the inside out” (from feature levels to logit levels).In Stage 1, we design a hierarchical fusion masking strategy, serving as a plug-and-play enhancement module. By horizontally fusing shallow texture features and deep semantic features in dermatoscopic images, it effectively guides the student in the first stage to reconstruct the feature maps used for prediction.In Stage 2, the student model undergoes a secondary distillation with the teacher model after updating its network parameters through Stage 1. A normalization operation is introduced during this process to enrich information for predicting labels and improve fitting.Extensive experiments are conducted on both the publicly available ISIC2019 dataset, which focuses on high-risk malignant skin cancer, and our private dataset featuring common skin diseases like acne. To further enhance model performance, we apply the RandAugment data augmentation method and a pseudo-shadow removal technique during dataset preprocessing.The effectiveness of our KD framework and the benefits brought about by its key components are significantly validated through comparisons with baseline models and other cutting-edge KD methods.

## 2. Related Work

### 2.1. Skin Disease Classification with CNN

Beginning with LeNet, successfully developed by Yann LeCun [14], an increasing number of researchers have turned their attention to the application of CNNs and backpropagation-based algorithms for establishing effective neural architectures in image classification tasks [15]. With the improvement of computational microprocessor capabilities, various CNN-based deep learning models have emerged, including AlexNet [16], Inception [17], VggNet [18], ResNet [19], DenseNet [20], EfficientNet [21], and ConvNext [22]. Considering the constraints of limited memory storage and minimal computational costs on some edge devices designed for specific tasks, lightweight convolutional neural networks such as SqueezeNet [23], MobileNet [24], and ShuffleNet [25] have been developed. These models require fewer parameters, enabling faster inference speeds, but this may come at the cost of compromises in overall model performance. In recent years, an increasing number of studies in the field of medical image classification for skin diseases have focused on addressing the challenge of balancing reliability and accuracy with model size. Hoang et al. introduced a novel variant of ShuffleNet, named wide-ShuffleNet, to determine the skin lesion type from the inputted segmentation results. Compared to other methods, this approach requires fewer parameters, yielding higher accuracy. Based on the HAM10000 dataset, it achieved an accuracy of 84.8%, and based on ISIC2019, it reached 82.56% [26]. Kumar et al. achieved a 12-fold reduction in parameter count by removing the top fire modules of the SqueezeNet architecture. They successfully applied this modified architecture to the HAM10000 dataset for skin cancer classification, demonstrating commendable performance results [27]. Liu et al. introduced the MFA-MViT method, a technique based on multi-level feature fusion and an attention mechanism. They incorporated an inverted residual block with channel attention (Attn-DWS module) to not only extract richer features but also achieve the weight distribution of features with a minimal number of parameters. The proposed method attained an accuracy of 92.56% on the ISIC2019 dataset, with a reduction of 16.16 M computations and 0.35 M parameters compared to the baseline network MobileViT [28]. Maqsood et al. presented a unified CAD model for skin lesion segmentation and classification using a deep learning framework. They extracted deep feature vectors from four modified CNN models (Xception, ResNet-50, ResNet-101, and VGG16), which were then combined through a convolutional sparse image decomposition fusion method. Their model achieved an impressive 93.47% accuracy based on the ISIC2019 dataset [29].

### 2.2. Knowledge Distillation

Broadly speaking, knowledge distillation refers to extracting knowledge from a larger deep neural network and transferring it into a smaller neural network model by aligning two different types of information sources [30]: (a) those utilizing the teacher’s output probabilities (logit-based). In 2015, Hinton et al. formally popularized vanilla knowledge distillation, with the core idea of employing the teacher’s prediction logits as soft labels to guide the student [9]. Zhou et al. addressed the bias–variance tradeoff by proposing weighted soft labels and validated its effectiveness through experiments with standard evaluation benchmarks [31]. Chi et al. introduced Normalized Knowledge Distillation (NormKD), aiming to customize the temperature for each sample based on the characteristic of the sample’s logit distribution. It demonstrated significantly better classification performance than vanilla knowledge distillation with almost no additional computational or storage costs [11]. (b) Those utilizing the teacher’s intermediate representations (feature-based). Romero et al. first introduced the idea that activations, neurons, or features from intermediate layers can serve as knowledge to guide the student model [32]. Inspired by this, Chen et al. enhanced the effectiveness of knowledge distillation by learning feature maps from the teacher network through two main techniques—shared classifier and generative adversarial network [12]. Kim et al. utilized a paraphraser to perform teacher factor extraction and transferred the learned factor to the student through convolutional operations [33]. Yang et al. proposed Masked Generative Distillation, a versatile feature-based distillation method applicable to diverse tasks. It involved masking random pixels of the student’s feature, compelling it to generate the teacher’s full feature [34].

In the human world, we integrate information from various sources to gain cognition. Similarly, in the task of knowledge distillation, a student model can also acquire multi-level knowledge. Based on this concept, most recent research has focused on multi-teacher distillation, using several distinct deep learning models as teachers to impart knowledge to the student. For instance, Liu et al. put forth an adaptive multi-teacher, multi-level knowledge distillation learning framework (AMTML-KD), which employed a straightforward multi-group hint strategy to learn distinct importance weights [35]. Additionally, Pham et al. presented a new framework integrating multi-teacher knowledge distillation and network quantization to learn low-bit-width DNNs. This approach facilitated collaborative learning among quantized teachers and mutual learning between these quantized teachers and the student [36]. In non-multi-teacher KD strategies, L. Li et al., building upon self-distillation, proposed a selective feature fusion module to enhance model performance and further compress the model [37]. C. Li et al. presented a novel online distillation method that integrated information and overall feature representations from all peer networks and the logits. This was achieved by embedding an ensemble and adaptive fusion branches between two parallel peer networks [38].

In the realm of intelligent classification tasks for medical skin diseases, research related to knowledge distillation remains scarce, with only a few notable advancements as follows. Wang et al. proposed SSD-KD, a method for skin disease classification that integrated diverse knowledge using a self-supervised dual-relationship knowledge distillation framework. This framework employed weighted softening of the output, enhancing the student model’s ability to extract richer knowledge from the teacher model. By employing EfficientNetB7 as the teacher model and EfficientNetB0 as the student model, the framework achieved an accuracy of 86.6% based on the ISIC2019 dataset [8]. Khan et al. trained a Distilled Student Network (DSNet) with approximately 260,000 parameters, achieving a 91.7% accuracy in detecting melanoma based on the ISIC dataset. DSNet’s parameter count was 15 times smaller than the minimal EfficientNet-B0. They further discovered that the proposed DSNet model performed optimally at a distillation temperature (T) of 10 [39]. Back et al. utilized an ensemble learning strategy to progressively learn from multiple teacher networks and trained a robust mobile deep neural network (DNN) for herpes zoster skin disease [40]. Experimental studies on knowledge distillation in various domains have highlighted its significant potential. This study aims to explore the application of knowledge distillation in dermatological image classification tasks. We believe that knowledge distillation possesses substantial untapped potential, particularly in the development of future mobile healthcare systems.

## 3. Proposed Framework: DSP-KD

### 3.1. Overall Structure

When there is a substantial disparity between the student and teacher, the student’s performance tends to decline. To maximize the extraction of information from the teacher and avoid the loss of valuable knowledge, a dual-stage progressive knowledge distillation (DSP-KD) framework is proposed (as depicted in Figure 2).

For each example in a given training set, it undergoes Stage 1, represented by the blue area in the figure. Different hierarchical representations—nuanced, coarse, and global—are extracted from the teacher. The Hierarchical Fusion (HF) module is employed to horizontally connect the top-down representations and generate attention feature maps. Guided by attention maps, masking is performed on the student to enable it to reconstruct the teacher’s extensive high-level feature map information. The distillation loss under the HF attention map is computed and fused with the original loss using a learnable parameter to obtain the final loss for the first stage, which is used for backward propagation to update the student model’s parameters.

In Stage 2, the student model with updated parameters undergoes secondary knowledge distillation with the original teacher. Unlike feature-based KD, logit-based KD is performed in Stage 2 to mimic the teacher’s predicted label information. To prevent the difficulty of distilling adequate information at a single set temperature, normalization operations are applied to each sample’s logit distribution. The differences between the distilled outputs of the student and teacher models vary between the two stages, with a larger gap in Stage 1 and a smaller one in Stage 2. Finally, the student model effectively enhances its performance by gradually aligning with the teacher, capturing various types of information, including shallow edge texture information, deep semantic information, advanced abstract complex information, and information related to the probability distribution at the output layer.

### 3.2. Methodology on Stage 1

#### 3.2.1. Masked Generative Distillation (MGD)

In Stage 1, we employ a feature-based distillation approach. The original method for distillation on features can be expressed as:(1)$$L_{orig}=\sum_{c=1}^{C}\sum_{h=1}^{H}\sum_{w=1}^{W}MSE(F_{c;h;w}^{T},f_{adapt}(F_{c;h;w}^{S}))$$
where $F^{T/S}$ represents the feature map extracted from the teacher or student network. $f_{adapt}$ is a learnable adaption function to align the student’s feature map $F^{S}$ with the teacher’s feature map $F^{T}$. MSE stands for the mean squared error, which denotes the fitting error between the teacher feature and the adapted student feature. C, H, and W denote the three dimensions of the feature map.

The approach of Formula (1) is to have the student directly mimic the teacher’s features. Building upon this, Yang et al. proposed a framework called Masked Generative Distillation (MGD) [34], which aims to assist the student in reconstructing features instead of simply imitating. This is achieved by masking random pixels of the student’s feature and compelling it to generate the teacher’s complete feature through a simple block. The distillation loss $L_{MGD}$ for MGD can be expressed as:(2)$$L_{MGD}\left(T,S\right)=\sum_{l=1}^{L}\sum_{c=1}^{C}\sum_{h=1}^{H}\sum_{w=1}^{W}MSE(T_{c;h;w}^{l},\xi (f_{adapt}\left(S_{c;h;w}^{l}\right)\cdot M_{h;w}^{l}))$$
where L represents the total number of distillation layers and C, H, and W denote the shape of the feature map. $T^{l}\in R^{C\times H\times W}$ and $S^{l}\in R^{C\times H\times W}(l=1,...,L)$ are the l-th feature maps of the teacher and student, respectively. $M_{h;w}^{l}$ denotes the l-th random mask to cover the student’s l-th feature. ξ(·) denotes the projector layer which includes two convolutional layers and one activation layer ReLU. At the end, we calculate the error loss between the recovered feature map and the teacher feature map.

#### 3.2.2. Hierarchical Fusion Masking Strategy

The feature mask regions are randomly generated in MGD, resulting in a lot of non-representative pixel usage. This causes the loss of crucial high-level details and the creation of regions without meaningful positions or information. In order to accurately improve the representation power of students’ features, a Hierarchical Fusion (HF) attention map is designed to derive corresponding masking maps, which exploit structural information and channel relationships for better feature representation. Feature maps at different levels are obtained during forward propagation by inputting images into the backbone, and they are extracted from multiscale receptive fields through average pooling operations. As shown in Figure 3, we upsample the attention maps generated by ${Level}_{n+1}$ and consider them as the activation of ${Level}_{n}$, with n∈{1,2,3}. The output of ${Level}_{n}$ first undergoes a 1 × 1 convolutional layer to reduce the channel dimension, and then is combined with the upsampled output vector of ${Level}_{n+1}$ through weighted summation after aligning dimensions.

By extensively leveraging local and global representations ranging from coarse to nuanced, this study addresses the limitation on Level 1, where shallow layers output large-sized feature maps that fail to fully exploit the information from channel attention mechanisms. The generated attention map exhibits robust semantic strength across all levels, including high-resolution tiers. We use $A_{h,w}^{HF}$ to denote the resulting HF attention map produced from the teacher feature, which is employed to generate the guided masking map $M^{HF}$ with the threshold φ, formulated as follows:(3)$$M_{h\subseteq \left[0,H\right),w\subseteq [0,W)}^{HF}=\left\{\begin{array}{l} 0,\;A_{h,w}^{HF}\geq \varphi \\ 1,\;Otherwise \end{array}\right.$$

After obtaining a composite of global, coarse, and nuanced masks representing teacher image features, we attempt to generate teacher feature maps by incorporating information from adjacent pixels located outside the mask. In contrast to MGD, our approach employs two 3 × 3 convolutional layers for the projector layer, along with a single activation layer using GELU instead of RELU. The structure of the generation block is illustrated in Figure 4.

By focusing on important areas in the teacher feature map, we selectively conceal features to enable the student model to reconstruct nuanced contextual information, effectively reducing distillation loss and improving overall performance. We denote the reconstructed feature map information for the student as $G_{HF}^{S}$, formulated as follows:(4)$$G_{HF}^{S}=\xi_{Conv}(f_{adapt}\left(F^{S}\right)\cdot M^{HF})$$
where $f_{adapt}$ is a learnable adaption function to align student’s feature map $F^{S}$ with teacher’s feature map $F^{T}$, while $\xi_{Conv}$ denotes the generation operation performed based on the HF attention map. Building upon the improved masked generation distillation method, the final loss calculation for Stage 1 is formulated as follows:(5)$$L_{HFfea}=\alpha \cdot \left[\sum_{c=1}^{C}\sum_{h=1}^{H}\sum_{w=1}^{W}MSE(G^{S},\;F^{T})\right]+L_{orig}$$
where C, H, and W denote the channel number, height and width of the feature map, respectively, $L_{orig}$ represents the initial feature-based distillation loss between the student and teacher, and we need a hyper-parameter α to balance distillation loss and original loss, leading to the final feature-based distillation loss under the HF attention map.

### 3.3. Methodology on Stage 2

#### Normalized Logits

In the vanilla knowledge distillation method, given C as the number of classes and a vector of logits $Z=\left[z_{1},\;z_{2},\;\ldots ,\;z_{i},\;\ldots ,\;z_{C}\right]\in R^{1\times C}$ as the outputs of the last fully connected layer of a deep model, the final prediction vector $P=\left[p_{1},\;p_{2},\;\ldots ,\;p_{i},\;\ldots ,\;p_{C}\right]\in R^{1\times C}$ can be expressed as [9]:(6)$$p\left(z_{i},T\right)=\frac{exp(z_{i}/T)}{\sum_{j=1}^{C}exp(z_{j}/T)}$$
where the temperature factor T is introduced to control the importance of each soft target. In most prior works, T was set as a fixed single value for the entire distillation process. Chi et al. conducted a sequence of experiments enabling the student to distill knowledge from labels with varying degrees of softening [11]. The results indicated a positive correlation between distillation performance and the frequency of temperatures within an appropriate range. To enable the student model to extract knowledge more comprehensively from each sample, the authors proposed Normalized Knowledge Distillation (NormKD), introducing a normalization operation before the SoftMax layer that approximates the logit distribution as a normal distribution with a scale parameter б. The final prediction vector $\tilde{P}=\left[\tilde{p_{1}},\;\tilde{p_{2}},\ldots ,\;\tilde{p_{i}},\;\ldots \;\tilde{p_{C}}\right]\in R^{1\times C}$ after normalization can be expressed as:(7)$$\tilde{p_{i}}=\frac{exp(z_{i}/(б\cdot T_{Norm}))}{\sum_{j=1}^{C}exp(z_{j}/(б\cdot T_{Norm}))}$$
where $T_{Norm}$ serves as an adaptive hyperparameter for scaling the normal distribution, derived from the distribution of logit-based losses under different temperatures. This approach incurs minimal additional computational or storage costs and can be readily applied to other logit-based KD methods. It addresses, to some extent, the unevenness issue associated with a single temperature, thereby enhancing the overall performance of the system.

After undergoing Stage 1 in our experiments, the student model acquires updated weight parameters. At this point, the student has effectively reconstructed the feature behaviors present in the teacher. Building upon the narrowed gap between the teacher and student, the student model enters a second round of normalized logit distillation with the teacher model. This process aims to enrich the information in predicting labels and enhance the fitting. Finally, the calculation of the expected distillation loss $L_{DSPkd}$ is as follows:(8)$$L_{DSPkd}=\frac{1}{N}\cdot \sum_{i=1}^{N}\left({\left(T_{Norm}\cdot {б}_{i}^{t}\right)}^{2}\cdot KLD({\tilde{P_{i}}}^{s},{\tilde{P_{i}}}^{t})\right)$$
where N is the number of samples and KLD is used to measure the difference between the final normalized probability distributions of the teacher and student models. Our knowledge combination method enhances logic and effectiveness, much like aiding a student with computer hardware knowledge to understand software systems, making comprehension for student model easier and more thorough.

## 4. Experiments and Results

### 4.1. Datasets and Data Preprocessing

Two datasets were utilized in our experiments: the International Skin Imaging Collaboration Challenge Dataset of 2019 (ISIC2019) [41] and a private skin disorders dataset (PSD-Dataset) provided by Peking Union Medical College Hospital, comprising 25,331 and 2900 dermatoscopic images, respectively. It is important to mention that all participants included in the PSD-Dataset provided informed consent.

The ISIC2019 dataset is a publicly available dataset aimed at improving the diagnosis and treatment of melanoma among eight different diagnostic groups: melanoma (MEL), melanocytic nevus (NV), basal cell carcinoma (BCC), actinic keratosis (AK), benign keratosis (solar lentigo/seborrheic keratosis/lichen planus-like keratosis) (BKL), dermatofibroma (DF), vascular lesions (VASCs), and squamous cell carcinoma (SCC). To further validate the generalization and practicality of our study, we also applied our DSP-KD framework to train the PSD-Dataset, which encompasses six common skin diseases: acne, melasma (ME), rosacea (ROS), discoid lupus erythematosus (DLE), Ota nevus (ON), and seborrheic dermatitis (SD).

We resized the ISIC2019 dataset to a size of 224 × 224 and the PSD-Dataset to a size of 512 × 512. Subsequently, both datasets addressed data imbalances by randomly augmenting samples in underrepresented classes through data augmentation techniques. Specifically, data preprocessing involved applying transformations to certain category datasets using 1–3 randomly generated operations from 14 different RandAugment methods [42]. The available transformations included identity, autoContrast, equalize, rotate, solarize, color, posterize, contrast, brightness, sharpness, shear-x, shear-y, translate-x, and translate-y. A hair removal version was generated for images with excessive noise. Figure 5 illustrates examples of the data transformations for the ISIC2019 dataset and the PSD-Dataset.

Following data preprocessing, the data were randomly divided into training and testing sets in an 8:2 ratio. Figure 6 and Figure 7, respectively, provide comprehensive details and sample images for each skin disease, including quantities before and after data augmentation, as well as the distribution between the training and testing sets for these two datasets.

### 4.2. Experimental Setup

For this work, all experiments were conducted with MMPretrain [43] and MMRazor [44] based on Pytorch 1.10.0, utilizing an NVIDIA GeForce RTX3060 GPU for accelerated computation. During single model training, the SGD optimizer was employed for 100 epochs, and during knowledge distillation from the teacher to the student model, training was extended to 120 epochs. For the initial 20 epochs, a linear learning rate scheduler was applied with an initial learning rate of 0.001. Starting from the 21st epoch, a cosine annealing learning rate scheduler was implemented, with a minimum learning rate set to 1 × 10^−5^, the momentum was 0.9, and the weight decay was 0.0005.

### 4.3. Evaluation Metrics

To facilitate a fair and comprehensive comparison of different CNNs and various knowledge distillation methods in the context of multi-class skin disease tasks, we employed four widely used metrics, namely accuracy, precision, specificity, and *F*1-score, which can be calculated using a useful tool, the confusion matrix. A confusion matrix is utilized to assess the performance of a classification algorithm by presenting a comprehensive breakdown of the predicted and actual class labels, as illustrated in Figure 8.

In Figure 8, True Positive (TP) and True Negative (TN) denote the number of subjects that are correctly predicted. False Positive (FP) and False Negative (FN) represent the number of subjects that are predicted incorrectly. Hence, the formulas for computing these four metrics are provided below.

Accuracy measures the overall correctness of the model by determining the ratio of accurately predicted instances to the total number of instances.
(9)$$Accuracy=\frac{TP+TN}{TP+FP+TN+FN}$$

Precision assesses the accuracy of positive predictions by calculating the ratio of true positive predictions to the total number of predicted positive instances.
(10)$$Precision=\frac{TP}{TP+FP}$$

Recall, also referred to as sensitivity or true positive rate, measures the model’s capability to identify all relevant instances. It determines the ratio of correctly predicted positive instances to the total number of actual positive instances.
(11)$$Recall=\frac{TP}{TP+FN}$$

Specificity evaluates the model’s ability to correctly identify negative instances. It calculates the ratio of accurately predicted negative instances to the total number of actual negative instances.
(12)$$Specificity=\frac{TN}{TN+FP}$$

F1-score is the harmonic mean of *precision* and *recall*. It provides a balanced measure between *precision* and *recall*.
(13)$$F1=\frac{2\times precision\times recall}{precision+recall}$$

To evaluate the classifier based on unbalanced data more accurately, the *macro-average* of the *F*1-score metric was also calculated, as shown in the Formula (14). This metric offers a comprehensive performance measure that treats each class equally. In our experiments, we determined the KD method’s superiority through overall accuracy, *F*1-score per class, and *macro-average* (*F*1-score).
(14)$$macro-average\;(F1)=\frac{1}{K}\sum_{i=1}^{K}{F1}_{i}$$
where K is the number of classes.

Furthermore, we measured each model’s weight in terms of the number of parameters (in millions) and inference time (s) and calculated the floating-point operations per second (FLOPs) to evaluate the computational complexity and practicality of skin lesion datasets, using a script provided by MMEngine [45].

### 4.4. Results

#### 4.4.1. Comparison of the Potential Teacher and Student Models

Before applying the proposed knowledge distillation framework, we conducted a comparative analysis of various mainstream convolutional neural networks of different scales based on both the ISIC2019 dataset and PSD-Dataset. We chose the optimal student model based on overall performance, model size, and computational speed to serve as the baseline for subsequent distillation experiments.

As shown in Table 1, the upper section lists three large networks with superior performance, considered as potential teacher models. The lower section comprises lightweight networks. The comparison reveals that teacher models indeed enhance the fundamental predictive ability but come with a more than tenfold increase in computational costs. In contrast, during student model training comparison, EfficientNet-B0 has five times more parameters than the most compact model among the four lightweight models. However, its accuracy is the lowest based on both the ISIC2019 dataset and the PSD-Dataset, reaching only 82.71% and 83.26%, respectively. Both ShuffleNetV2 and MobileNetV2 perform comparably, with only a 0.13% difference in accuracy on the ISIC2019 dataset, but MobileNetV2 exhibits a noticeable increase in computational complexity and parameters. Overall, ShuffleNetV2 achieves the best skin disease classification performance within the minimal computational cost and memory requirements. Therefore, ShuffleNetV2 was designated as the selected student model.

#### 4.4.2. Overall Comparison of KD Methods with Different Knowledge Sources

To validate the effectiveness and practicality of our distillation framework, we compared different knowledge distillation methods with various knowledge sources: absorbing only logits, absorbing only features, and state-of-the-art multi-source knowledge fusion frameworks. All methods were implemented within the same execution framework, with the student ShuffleNetV2 sequentially learning from the teacher models Res2Net101, ConvNext, and EfficientNet-B4. To ensure the credibility of table’s results, we conducted each experiment multiple times, with a minimum of three trials under identical conditions. Subsequently, we computed the standard deviation for all the metrics obtained. The standard deviation values consistently remained below 0.01. Table 2 presents the accuracy results of ShuffleNet-V2 under various KD methods based on the ISIC2019 dataset.

It can be observed that, since both the selected teacher models and the student model are convolutional neural networks with similar structures, all distillation methods more or less improved the student’s performance. As shown in Table 2, based on the ISIC2019 skin disease dataset, gains from learning only logits or only features were not significant, but when using diverse knowledge sources, the student achieved an improvement of 1.71% to 3.74%. Notably, our method, employing Res2Net101 as the teacher, significantly boosted ShuffleNetV2’s classification capability from 86.21% to 92.83% in multi-class skin disease, surpassing the teacher model by 0.24%. This is remarkable, as it uses fewer parameters and requires less time for training and prediction. In the case of the other two teacher models, classification performance also improved by 3.18% and 2.8%, respectively. This effectively reveals the generalization ability of our method. Figure 9 depicts the variation in training accuracy for three sets of teacher–student pairs under the DSP-KD distillation framework.

Similarly, we employed the same experimental strategy using the private skin disorders dataset, and the corresponding accuracy results are presented in Table 3. As shown in Table 3, we observed that, in comparison to ISIC2019, there is not a significant difference in the classification accuracy between undistilled teacher and student models based on the PSD-Dataset. This suggests that lightweight CNNs can exhibit commendable performance with small datasets. However, this also poses a challenge to achieve significant improvements when distilling knowledge into the student. When only incorporating knowledge from logits, there was no substantial enhancement in the student’s performance. In the experiment with EfficientNet-B4/ShuffleNetV2, the distilled ShuffleNetV2 even showed a decrease of 0.45% compared to the undistilled counterpart. This aligns with other research conclusions, emphasizing the challenge of improving classification tasks through pure logit-based KD due to structural differences between teachers and students [46]. When performing experiments on multi-source knowledge fusion distillation methods, the accuracy of the distilled student model approaches that of the teacher. Upon learning from the optimal teacher Res2Net101, the SSD-KD framework achieved an accuracy of 93.81%, while for our pro-posed DSP-KD, it reached 94.93%. In summary, our framework outperforms the optimal fusion knowledge distillation framework by 1.12% and surpasses the original teacher and student models by 0.27% and 5.45%, respectively.

The above results indicate that our method outperforms others based on both datasets in terms of accuracy. Meanwhile, it can transfer knowledge across CNN models with different internal structures, whether based on EfficientNet-B4/ShuffleNetV2, ConvNext/ShuffleNetV2, or Res2Net101/ShuffleNetV2.

#### 4.4.3. Per-Class Performance Comparison of KD Methods

To examine whether superior performance is exclusive to the majority class, we further compared the independent performance of different KD methods (representatives from the previous experiments) for each class. Since the Res2Net101/ShuffleNetV2 pair consistently achieved the highest overall performance, experiments in this section were based on this collaboration. Table 4 and Table 5 present the *F*1-score comparison of different knowledge sources’ distillation methods with each class and calculate the macro-average.

Based on both datasets, our method attains the highest *macro-average* (*F*1). Specifically, for melanocytic nevus (NV) of the ISIC2019 dataset, DSP-KD outperforms another diverse-source distillation framework, SSD-KD, by 4.88%, while for the dataset DF, which has the least amount of data, DSP-KD achieves a notable 10.51% improvement. This substantial improvement indicates that our method excels not only in categories with more data but also in those with fewer data. We observed a similar performance trend in the PSD-Dataset. Due to the similar pinkish clustered appearance of rosacea and acne, it is challenging to significantly boost classification performance by simply restoring shallow texture features from the teacher. However, with the implementation of DSP-KD, we reinforced structural information and channel relationships to improve feature representation. As a result, we observed a 1.41% enhancement for acne and a 4.57% improvement for rosacea (ROS), respectively. Figure 10 illustrates the experimental results in the form of a confusion matrix, providing a detailed showcase of our method’s predictive capabilities in each category.

#### 4.4.4. Time Inference

As demonstrated in Table 6, we take a closer look at the computational aspects represented by parameters and inference time. Based on the Res2Net101/ShuffleNetV2 pair, our distillation method significantly boosts the student model’s accuracy by an additional 6.62%, while reducing 0.172 million parameters along with a 0.023 decrease in inference time. Additionally, whether comparing with another state-of-the-art knowledge distillation framework, SSD-KD, or a lightweight model focused on skin disease prediction, wide-ShuffleNet, the outcomes once again confirm the efficiency of our proposed DSP-KD method with fewer parameters and faster speed.

### 4.5. Ablation Study of DSP-KD

DSP-KD is a progressive distillation framework that integrates two distinct knowledge distillation methods. Therefore, it is essential to explore the roles and impacts of different components within the entire framework. We conducted an ablation study based on the publicly available ISIC2019 dataset, with ShuffleNetV2 as the student model, which also serves as our enhanced baseline. As evident from Table 7, the HF masking strategy introduced in Stage 1 enhances performance by 2.35% over MGD. In Stage 2, the replacement of the logit-based distillation method surpasses the traditional Vanilla-KD by an additional 0.89%. Combining the progressive stages of Stage 1 and Stage 2 outperforms no knowledge fusion by 1.94%. This demonstrates the effectiveness and necessity of our framework using a progressive approach and improved knowledge distillation strategies.

## 5. Conclusions and Future Work

In this paper, we proposed a novel diverse-source knowledge fusion distillation framework, named DSP-KD. This framework employs a dual-stage progressive distillation approach, where the improved first-stage feature distillation results guide the second-stage logit-based distillation. The aim is to (1) progressively narrow the gap between the teacher and student to enhance the distillation effect (2) and maximize effective knowledge transfer, allowing the teacher to guide the student from the inside out. In Stage 1, an HF masking strategy based on feature distillation is introduced to utilize feature maps containing different hierarchical representations of the teacher to guide the reconstruction of crucial regions of interest in the student’s feature maps. In Stage 2, the concept of normalized logits is introduced to enhance the overall prediction capability of the student. By using ShuffleNetV2 as the student model, we perform distillation experiments with three CNNs that excel in the current classification tasks. The entire distillation experiment conducted through DSP-KD demonstrates superior results based on both the public dataset ISIC2019 and our private skin disorders dataset. The distilled ShuffleNetV2 shows improved predictive performance while maintaining low model complexity and storage requirements. As hypothesized, improving knowledge absorption strategies can further enhance the representational capability of lightweight student models across various skin disease classification tasks. However, allowing the student model to explore knowledge freely during training is unreliable, and the purposeful and progressive feeding of knowledge is necessary during distillation, as demonstrated in our proposed framework.

The selection of the teacher model and differences in the teacher–student structure can impact distillation performance, motivating the progressive distillation approach proposed in this study. Currently, most knowledge distillation methods only consider distillation within the same architecture. The DSP-KD framework presented in this paper is designed specifically for distilling knowledge from one CNN to another. Future work will focus on refining the framework to apply it to cross-architecture teacher–student models, enabling students to learn more comprehensive and complementary information. Extracting knowledge across different architectural frameworks poses challenges, and optimizing the “transferability” of heterogeneous distillation frameworks is an ongoing research direction.

## Figures and Tables

**Figure 1 bioengineering-11-00070-f001:**
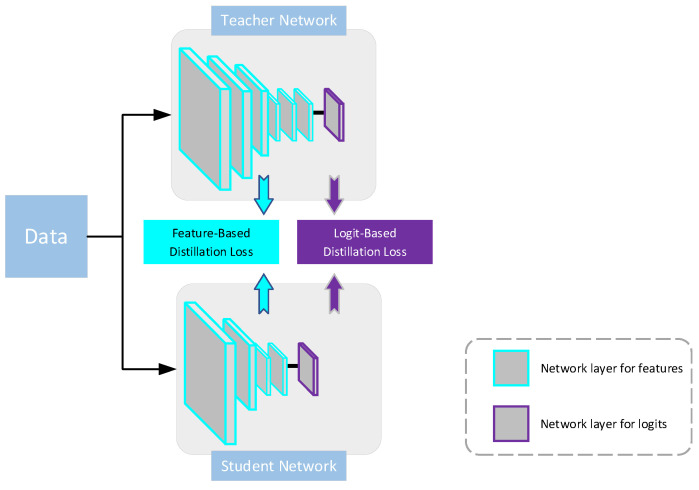
The two knowledge sources obtained from the teacher network.

**Figure 2 bioengineering-11-00070-f002:**
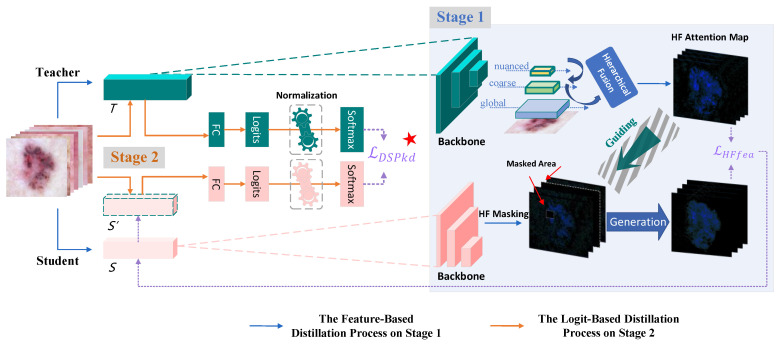
The overall structure of the proposed DSP-KD framework.

**Figure 3 bioengineering-11-00070-f003:**
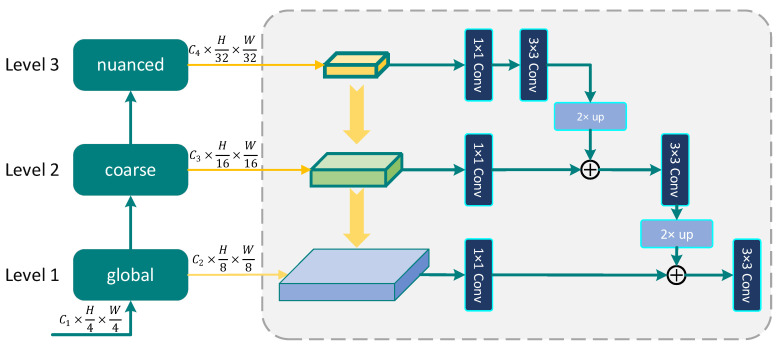
The structure of the proposed HF attention map mechanism.

**Figure 4 bioengineering-11-00070-f004:**
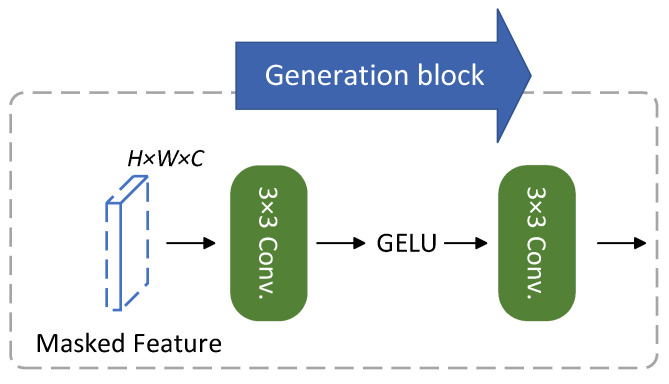
Illustration of the generation block.

**Figure 5 bioengineering-11-00070-f005:**
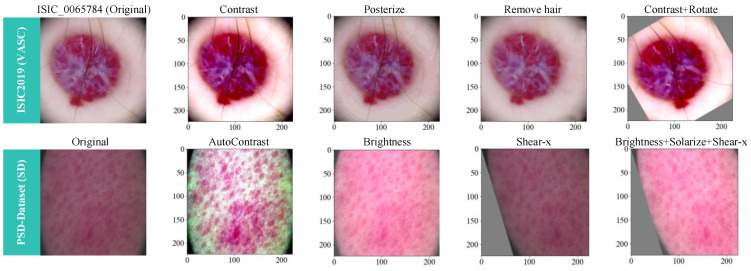
Demonstration of the effects of data augmentation techniques.

**Figure 6 bioengineering-11-00070-f006:**
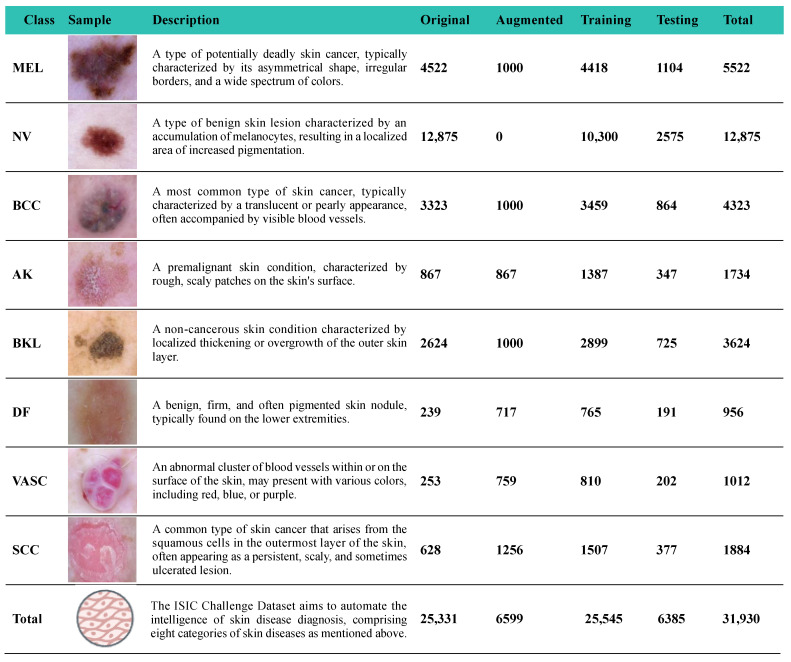
Sample images along with the detailed introduction about the ISIC2019 dataset.

**Figure 7 bioengineering-11-00070-f007:**
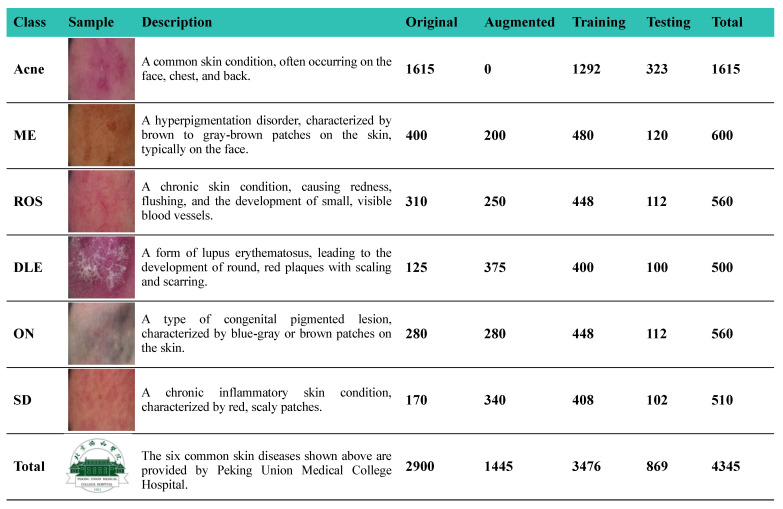
Sample images along with the detailed introduction about the PSD-Dataset.

**Figure 8 bioengineering-11-00070-f008:**
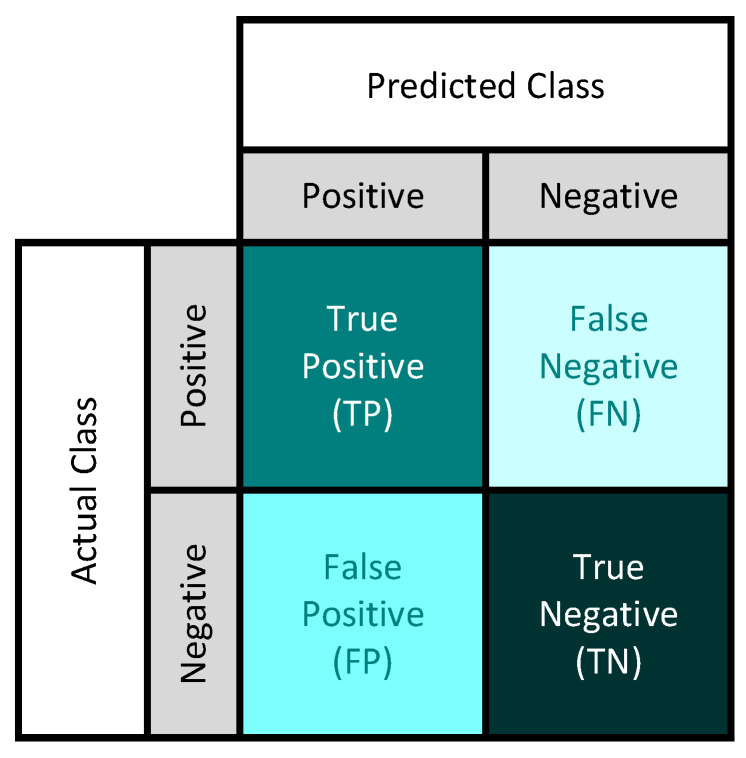
Confusion matrix.

**Figure 9 bioengineering-11-00070-f009:**
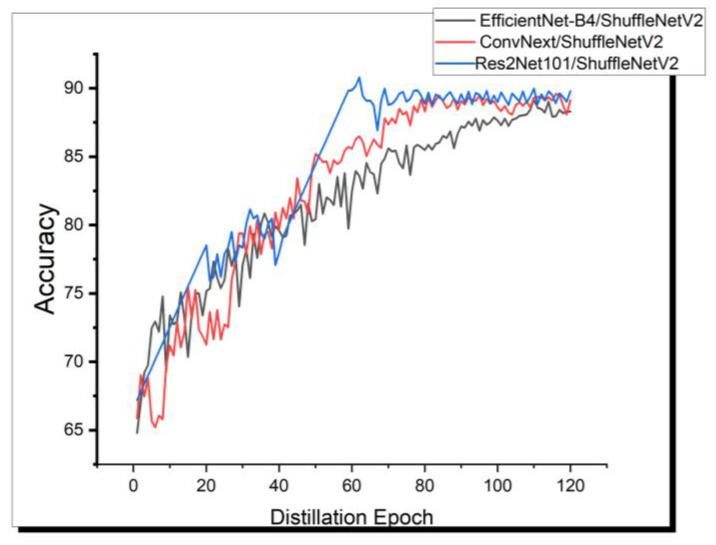
The training process of the three model pairs (Res2Net101/ShuffleNetV2, ConvNext/ShuffleNetV2, EfficientNet-B4/ShuffleNetV2) under our DSP-KD framework.

**Figure 10 bioengineering-11-00070-f010:**
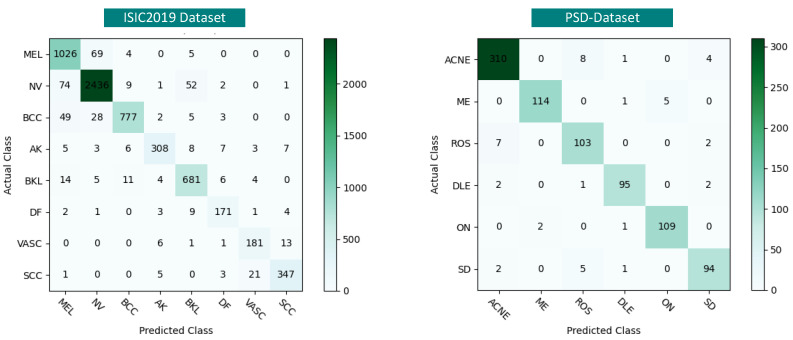
The confusion matrix of the training results of our proposed DSP-KD method.

**Table 1 bioengineering-11-00070-t001:** Comparison of performance and computation costs based on the ISIC2019 dataset and the PSD-Dataset.

Model	Parameters /Million	FLOPs /Giga	Accuracy/%		Precision/%		Specificity/%		*F*1-Score/%	
			ISIC2019	PSD-Dataset	ISIC2019	PSD-Dataset	ISIC2019	PSD-Dataset	ISIC2019	PSD-Dataset
Res2Net101	43.174	8.102	92.59	94.66	91.96	94.26	98.08	99.35	88.67	85.24
ConvNeXt	87.575	15.383	90.21	92.18	90.64	91.98	97.25	98.17	87.45	81.15
EfficientNet-B4	19.342	1.543	89.54	91.23	88.72	92.21	97.93	98.24	87.22	82.66
ShuffleNetV1	0.912	0.143	85.37	86.14	85.11	86.97	90.63	91.45	79.63	77.32
ShuffleNetV2	1.262	0.148	86.21	89.48	86.56	90.30	92.40	95.24	82.49	80.71
EfficientNet-B0	5.289	0.4	82.71	83.26	82.19	82.88	88.35	88.93	76.85	75.95
MobileNetV2	3.505	0.313	86.34	88.94	86.32	88.64	91.26	96.34	81.30	80.22

**Table 2 bioengineering-11-00070-t002:** Accuracy comparison: our method vs. other KD methods for the Res2Net101/ShuffleNetV2, ConvNext/ShuffleNetV2, EfficientNet-B4/ShuffleNetV2 model pairs in the ISIC2019 dataset.

Teacher model		Res2Net101	ConvNext	EfficientNet-B4
Student model		ShuffleNetV2	ShuffleNetV2	ShuffleNetV2
Teacher accuracy		92.59	90.21	89.54
Student accuracy		86.21	86.21	86.21
**Knowledge source**	**Distillation Method**	**Accuracy/%**	**Accuracy/%**	**Accuracy/%**
Logits	Vanilla-KD (Hinton et al.) [9]	87.39	86.98	87.20
Logits	WSLD (Zhou et al.) [31]	87.77	86.83	87.38
Features	Fitnets (Romero et al.) [32]	88.13	87.55	86.85
Features	MGD (Yang et al.) [34]	88.54	88.01	87.92
Diverse	EML (Li et al.) [38]	87.92	88.95	88.34
Diverse	SSD-KD (Wang et al.) [8]	89.95	88.23	88.12
Diverse	DSP-KD (Ours)	92.83	89.39	89.01

**Table 3 bioengineering-11-00070-t003:** Accuracy comparison: our method vs. other KD methods for the Res2Net101/ShuffleNetV2, ConvNext/ShuffleNetV2, EfficientNet-B4/ShuffleNetV2 model pairs in the PSD-Dataset.

Teacher model		Res2Net101	ConvNext	EfficientNet-B4
Student model		ShuffleNetV2	ShuffleNetV2	ShuffleNetV2
Teacher accuracy		94.66	92.18	91.23
Student accuracy		89.48	89.48	89.48
**Knowledge source**	**Distillation Method**	**Accuracy/%**	**Accuracy/%**	**Accuracy/%**
Logits	Vanilla-KD (Hinton et al.) [9]	90.11	89.93	89.03
Logits	WSLD (Zhou et al.) [31]	90.57	90.19	89.46
Features	Fitnets (Romero et al.) [32]	92.36	90.64	89.96
Features	MGD (Yang et al.) [34]	91.29	91.08	90.23
Diverse	EML (Li et al.) [38]	92.03	91.37	90.74
Diverse	SSD-KD (Wang et al.) [8]	93.81	92.70	91.39
Diverse	DSP-KD (Ours)	94.93	92.89	92.01

**Table 4 bioengineering-11-00070-t004:** Per-class *F*1-score comparison: our method vs. other KD methods in the ISIC2019 dataset, using Res2Net101 as the teacher model and ShuffleNetV2 as the student model.

KD Method	MEL	NV	BCC	AK	BKL	DF	VASC	SCC	*Macro-Average* (*F*1)
Vanilla-KD (Hinton et al.) [9]	86.31	86.99	79.13	74.04	78.56	71.24	75.91	73.40	78.19
Fitnets (Romero et al.) [32]	86.06	87.58	86.96	78.37	81.67	75.21	74.30	82.44	81.57
SSD-KD (Wang et al.) [8]	87.69	90.33	88.21	86.14	85.87	78.55	83.07	88.32	86.02
DSP-KD (Ours)	90.20	95.21	93.00	91.12	91.66	89.06	87.86	92.66	91.35

**Table 5 bioengineering-11-00070-t005:** Per-class *F*1-score comparison: our method vs. other KD methods in the PSD-Dataset, using Res2Net101 as the teacher model and ShuffleNetV2 as the student model.

KD Method	ACNE	ME	ROS	DLE	ON	SD	*Macro-Average* (*F*1)
Vanilla-KD (Hinton et al.) [9]	93.35	87.46	79.31	81.22	84.97	80.07	84.40
Fitnets (Romero et al.) [32]	93.99	89.06	80.32	84.97	85.84	86.14	86.72
SSD-KD (Wang et al.) [8]	94.86	95.10	85.39	89.62	92.27	86.44	90.61
DSP-KD (Ours)	96.27	96.61	89.96	95.48	96.46	92.16	94.49

**Table 6 bioengineering-11-00070-t006:** Parameter numbers and inference time in the ISIC2019 dataset.

Method	Top-1	Parameters /Million	Inference Time/s
Res2Net101(T)	92.59	43.174	0.297
ShuffleNetV2(S)	86.21	1.262	0.046
wide-ShuffleNet [26]	85.48	1.8	0.101
T/S+SSD-KD	89.95	1.13	0.072
T/S+DSP-KD	92.83	1.09	0.023

**Table 7 bioengineering-11-00070-t007:** Ablation study results based on the ISIC2019 dataset.

Method	Top-1	Δ
ShuffleNetV2(S)	86.21	-
S + MGD (a)	88.54	S + 2.33
S + MGD + HF (Stage 1)	90.89	a + 2.35
S + VKD (b)	87.39	S + 1.18
S + NKD (Stage 2)	88.28	b + 0.89
S + MGD + HF + VKD	91.47	Stage 1+0.58
S+MGD + HF + NKD (DSP-KD, Ours)	92.83	Stage 1 + 1.94

## Data Availability

We analyzed a public dataset in this study. It is available at https://challenge.isic-archive.com/data/#2019 (accessed on 15 November 2023).

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
