# Peer review of "DSP-KD: Dual-Stage Progressive Knowledge Distillation for Skin Disease Classification"

_bioengineering, 2024, doi:10.3390/bioengineering11010070_

Round 1

Reviewer 1 Report

Comments and Suggestions for Authors

- The authors used in their experiments a "private dataset" provided by a local Hospital and no reference is made to its approval by the healthcare unit's Ethics Committee (and/or equivalent/other regulatory bodies); that should be clarified;

- Despite augmentation, the  datasets are still highly imbalanced, so it highly recommended to have assessment metrics per class (ex:  mean Average Precision- mAP); 

- Since the the models' dimensions is a key motivation for the work, it would be relevant to benchmark model sizes (in MB) as part of the computation burden;

- Performance results are often close at less than 1%... is it possible to prove estimates for the associated uncertainties?

Comments on the Quality of English Language

- Several issues were raised above

Reviewer 2 Report

Comments and Suggestions for Authors

The manuscript introduces a knowledge distillation framework, DSP-KD, employing a dual-stage progressive distillation approach. The experimental results on ISIC2019 and a private dataset showcase superior performance of the distilled ShuffleNetV2 model in skin disease classification tasks. With that said, I would like to provide the following comments on the manuscript:

1) Overall, the manuscript is well written. However, attention is needed in refining the AI-generated sentences, for example in lines 83-85. A thorough review and revision of these type of sentences are required. 

2) Long sentence, such as this one, is very difficult to follow : "Unlike other fusion knowledge distillation frameworks that commonly adopt a multi-teacher simultaneous distillation approach, our framework employs a dual-stage distillation with the same teacher, gradually minimizing the adverse impact of teacher-student differences on distillation effectiveness, and maximizing the learning of the teacher's information "from the inside out" (from feature levels to logits levels)". Kindly review the paper and rectify any lengthy sentences for improved readability.

3) Kindly append a column to Table 2, incorporating F1-score data. The utilization of the F1-score is deemed more suitable than the accuracy metric, given the inclusion of data augmentation in the database.

4) The manuscript could benefit from a more in-depth discussion on potential limitations and challenges faced during the experimentation process.

Comments on the Quality of English Language

Minor editing of English language required

Reviewer 3 Report

Comments and Suggestions for Authors

The work of the authors is interesting and worth consideration. As a clinician, I would like to read in the final part more discussion on the potential applications of the model, and a comparison with similar model from a clinical/practical point of view, e.g.: does the model require less time/energy/bandwidth to be run? Are there already clinical applications?

Moreover, how do you explain the differences in accuracy between the public dataset and your private one? Were the images in your dataset somehow different, and how?

Then , concerning diagnoses, there were differences in diagnosing single entities? I mean, was the model more accurate in diagnosing BCC or melanoma or SCC or AK and so on? For a clinical point of view this could be more interesting.

Round 2

Reviewer 1 Report

Comments and Suggestions for Authors

I acknowledge the attention to previous comments, correctly addressing the  issues raised